# Automated segmentation of organs and tumors from partially labeled 3D CT in MICCAI FLARE 2023 Challenge

Andriy Myronenko[0000−0001−8713−7031], Dong Yang[0000−0002−5031−4337], Yufan He[0000−0003−4095−9104], and Daguang Xu[0000−0002−4621−881X]

NVIDIA
amyronenko@nvidia.com

**Abstract.** Automated organ and tumor segmentation from 3D CT is one of the key tasks in medical image analysis. In this work, we describe our solution to the FLARE 2023[1] challenge (team NVAUTO). We use an automated segmentation method Auto3DSeg[2] available in MONAI[3]. Our method achieves a 93% average organs class Dice score, and a 43% tumor class Dice score based on the 5-fold cross validation.

**Keywords:** Auto3DSeg · MONAI · Segmentation.

## 1  Introduction

Three dimensional computer tomography (CT) is one of the key medical imaging modalities, which gives insights into the human body anatomy and has many applications in disease detection and monitoring. Automated 3D segmentation of organs and tumors from 3D CT is a valuable tool for treatment planning and disease analysis. Deep learning methods are able to learn from image examples, and can be run fast in clinical practice. The accuracy of such methods depends on the ground truth data available for training. The amount of public 3D CT labeled data remains low, since it is challenging and time consuming to create ground truth 3D labels, and requires a trained physician or radiologist expertise.

Fast, Low-resource, and Accurate oRgan and Pan-cancer sEgmentation in Abdomen CT (FLARE23) challenge aims to promote the development of universal organ and tumor segmentation in 3D CT scans [14,15]. This year, FLARE23 combines several publicly available datasets to achieve one of the largest 3D CT datasets, albeit with only partial labels. FLARE23 includes a large subset of cases with tumor only labels in various locations (without focusing on a specific area). Finally a large unlabeled CT subset is also provided to facilitate potentially unsupervised learning. In total, FLARE23 includes 4000 CT cases: 2200 partially labeled, and 1800 unlabeled (see Sec. 2.3 for more details). Such

---

[1] https://codalab.lisn.upsaclay.fr/competitions/12239
[2] https://monai.io/apps/auto3dseg
[3] https://github.com/Project-MONAI/MONAI

a large CT data collection provides diverse variability of data examples, which come from a variety of medical institutions and populations. At the same time it poses a big algorithmic challenge, since ground truth labels are only partial, contoured by different people with different protocols. FLARE23 also adds another layer of complexity with a requirement for low GPU memory and short time processing, with an ideal setting of segmenting image under 15 seconds and under 4GB peak GPU memory.

Our solution is based on the automated supervised training of Auto3DSeg from MONAI [2]. We re-use supervised semantic segmentation where possible, and convert the data to a form acceptable for supervised training. Specifically we re-label the missing labels in each image using our own pseudo-labels, and retrain on the full dataset as it were a supervised problem. In a nutshell, our submission is an ensemble of 5 models (SegResNet [18] from MONAI[4]) processed at $1 \times 1 \times 1\text{mm}^3$ CT resolution. We made several optimizations to the inference pipeline to be able run end-to-end inference at ∼3.6GB peak GPU memory and ∼25 seconds per CT image on average. We describe the method and the optimizations in Sec. 2.

## 2   Method

We implemented our approach with MONAI [2] using Auto3DSeg open-source project. Auto3DSeg is an automated solution for 3D medical image segmentation, utilizing open source components in MONAI, offering both beginner and advanced researchers the means to effectively develop and deploy high-performing segmentation algorithms.

The labeled portion of the FLARE23 dataset are only partially labeled, with many cases including only a single labeled class. Since Auto3DSeg is a fully supervised segmentation solution, we split training into two stages: a) training on the fully labeled small subset (we found 250 cases out of 2200 to include all labels) b) pseudo-labeling the missing classes in the rest of the cases, and re-training a second round (second supervised training). Pseudo-labeling is common practice for semi-supervised learning, which was also used in the previous year FLARE22 champion solutions [10,22]. The unlabeled portion of the FLARE23 dataset was not used (see Sec. 2.3 for more details).

The fully supervised segmentation training with Auto3DSeg is simple:

```bash
#!/bin/bash
python -m monai.apps.auto3dseg AutoRunner run \
    --input="./input.yaml --algos=segresnet"
```

where a user provided input configuration (input.yaml) including only a few lines:

```yaml
# This is the YAML file "input.yaml"
modality: CT
```

---

[4] https://docs.monai.io/en/latest/networks.html#segresnetds

```
3 datalist: "./dataset.json"
4 dataroot: "/data/flare23"
```

When running this command, Auto3DSeg will analyze the dataset, generate hyperparameter configurations for several supported algorithms, train them, and produce inference and ensemble. The system will automatically scale to all available GPUs and also supports multi-node training. The 3 minimum user options (in input.yaml) are data modality (CT in this case), location of the dataset (dataroot), and the list of input filenames with an associated fold number (dataset.json). We generate the 5-fold equal split assignments randomly (one based on the fully labeled 250 cases, and the second one based on the pseudo-labeled 2200 cases)

Currently, the default Auto3DSeg setting trains three 3D segmentation algorithms: SegResNet [18], DiNTS [7] and SwinUNETR [6] with their unique training recipes. SegResNet and DiNTS are convolutional neural network (CNN) based architectures, whereas SwinUNETR is based on transformers [21]. Here we used only SegResNet for simplicity and describe its training procedure in this paper to be self-inclusive. At inference, we ensemble 5 best model checkpoints of SegResNet (5-folds). Since FLARE23 specifically required a fast and low GPU memory inference, we made several trade offs between accuracy and compute time, which we describe in Sec. 2.5

### 2.1   Preprocessing

- we resample data to $1 \times 1 \times 1\text{mm}^3$ isotropic resolution using tri-linear interpolation for CT images, and nearest neighbor interpolation for label images.
- we normalize images to $[0, 1]$ intensity interval from a $[-250, 250]$ input CT interval.
- for the first training stage only the 250 fully labeled images are used (based on the label analysis in Sec. 2.3)

Image resampling and normalization are done on the fly during training, and are a part of the training processing (in contrast to an off-line preprocessing). Auto3DSeg also caches in RAM the resampled data during the first training epoch, to speed up training automatically. If the RAM size is not sufficient, only a fraction of the data is cached, and the rest is recomputed at each epoch. This allowed us to avoid an off-line resaving step of the resampled data (which is quite large for FLARE23 dataset), and to quickly experiment with different re-sampling strategies (e.g. to try different image resolutions).

### 2.2   Proposed Method

The underlying network architecture is SegResNet [18] from MONAI[5]. It is an asymmetric encode-decoder based semantic segmentation network. It is a U-net alike convolutional neural network with deep supervision (see Figure 1).

---

[5] https://docs.monai.io/en/latest/networks.html#segresnetds

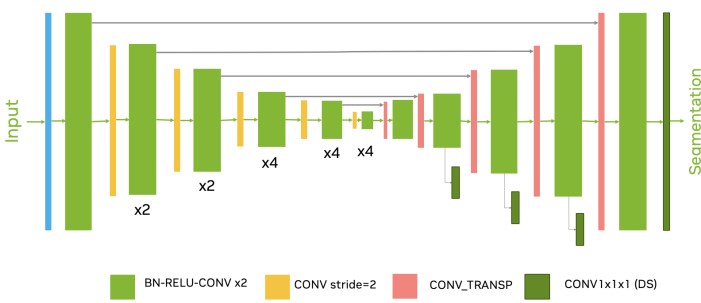

**Fig. 1.** SegResNet network configuration. The network uses repeated ResNet blocks with batch normalization and deep supervision

The encoder part uses residual network blocks, and includes 5 stages of 1, 2, 2, 4, 4 blocks respectively. It follows a common CNN approach to downsize image dimensions by 2 progressively and simultaneously increase feature size by 2. All convolutions are $3 \times 3 \times 3$ with an initial number of filters equal to 32. The decoder structure is similar to the encoder one, but with a single block per each spatial level. Each decoder level begins with upsizing with transposed convolution: reducing the number of features by a factor of 2 and doubling the spatial dimension, followed by the addition of encoder output of the equivalent spatial level.

We use a combined Dice-Focal loss[17,12][6], and sum it over all deep-supervision sublevels[7]:

$$Loss = \sum_{i=0}^{4} \frac{1}{2^i} Loss(pred, target^{\downarrow}) \tag{1}$$

where the weight $\frac{1}{2^i}$ is smaller for each sublevel (smaller image size) $i$. The target labels are downsized (if necessary) to match the corresponding output size using nearest neighbor interpolation.

We use the AdamW optimizer with an initial learning rate of $2e^{-4}$ and decrease it to zero at the end of the final epoch using the Cosine annealing scheduler[8] with 3 warmup epochs. We use batch size of 1 (per GPU), random crop of $224{\times}224{\times}224$, weight decay of $1e^{-5}$, and optimize for 300 epochs. We use several augmentations including random rotation and scale (in axial plane only), random flips, random histogram shift and random contrast adjustment.

The same exact network architecture and schedule was used both in the first stage training on the fully labeled data (250 cases) and then in the second stage on the pseudo-labeled data (2200 cases).

---

[6] https://docs.monai.io/en/stable/losses.html#dicefocalloss

[7] https://github.com/Project-MONAI/MONAI/blob/dev/monai/losses/ds_loss.py

[8] https://docs.monai.io/en/latest/optimizers.html#warmupcosineschedule

### 2.3   Partial labels

We follow a common practice of pseudo-labeling to tackle the partially labeled data. Auto3DSeg initial step uses DataAnalyzer() from MONAI to create a compact data description of the available data statistics (see data_stats_by_case.yaml file in the working directory). Among other things it will summarize available labeled classes per case, which is a convenient way to explore the data. Table 1 shows various data subsets found in the overall FLARE23 data.

**Table 1.** FLARE23 analysis of data subsets.

| Number of cases | Number labeled classes |
| --- | --- |
| 250 | 14 labeled classes: liver, right kidney, spleen, pancreas, aorta inferior vena cava, right adrenal gland, left adrenal gland, gallbladder, esophagus, stomach, duodenum, left kidney, tumor. |
| 1062 | 6 labeled classes: liver, right kidney, spleen, pancreas, left kidney, tumor. |
| 888 | 1 labeled class: tumor. |
| 1800 | unlabeled (unused in our method) |

The inhomogeneity of the labeled cases comes from the inhomogeneity of various datasets used to assemble the overall FLARE23 data.

We train a fully supervised segmentation model (5-folds) on the fully labeled subset of 250 cases. Then, we run inference ensemble of 5 models on the remaining partially labeled cases and assign the pseudo-labels to the missing classes (only the missing labels were replaced, and the original partial label subsets were preserved). One exception is the tumor class, we did not add it as a pseudo-label (and maintain the original tumor labels for all data cases). This allowed us to have all 2200 cases fully labeled, and used them for a second round of supervised segmentation training. We did not use unlabeled cases in our method

We did experiment initially with bypassing pseudo-labeling and training on the input 2200 partially labeled cases as is. Those experiments were futile, as the network (surprisingly accurately) learned to predict only partial labels (on the partially labeled training data), and full labels (on the fully labeled data), after all, those are the examples it learned from.

### 2.4   Unlabeled images.

Unlabeled images were not used. The provided pseudo labels generated by the previous year FLARE22 top algorithms [10,22] were not used.

### 2.5   Inference

Our inference is an ensemble of 5 SegResNet model checkpoints, each inferred with a sliding window strategy (ROI size 224×224×224) over a resampled image to $1 \times 1 \times 1\text{mm}^3$ resolution. Initially we attempted to ensemble more model checkpoints, but even for the 5 models, the inference time were prohibitively long. Below we list some of the main techniques we used to speed up the overall inference.

**Global abdominal region cropping.**   Some of the input CT images are full body CT, as large as 512×512×2000 voxels, with abdominal organs occupying only a portion of the image. To reduce image size, we trained a separate simple binary segmentation network to segment 1 class (a union of all abdominal organs) at $4 \times 4 \times 4\text{mm}^3$ low resolution. We used SegResNet with 1,2,2,4 blocks and only 16 initial filters, and trained it on 250 fully labeled cases with 128×128×128 ROI. At $4 \times 4 \times 4\text{mm}^3$ resolution, even large CT images can be inferred fast. The segmentation accuracy of the network was not essential since we wanted only to detect an approximate abdominal region. A dedicated bounding box detector would have been faster, but here we re-used the existing Auto3DSeg functionality. After running this network on the input image, we crop it to the abdominal region to reduce the image size for the subsequent main network inference/ensemble. We crop the input image only in axial (x,y) planes, but keep the full inferior-superior length of the CT. This is because the tumor class is potentially present anywhere in the body (including head and neck region or legs) and is not conforming to the abdominal region alone in FLARE23 data. Nevertheless with this technique we significantly reduced the input image size.

**First model cropping.** To further reduce the computational time, we further crop the input CT based on the result of the very first model (first out of 5 in the ensemble). If the first model did not detect tumors outside of the abdominal region, the inferior-superior region is cropped to further reduce the input image size (which is especially effective for full body 3D CT). This technique, however, is a trade off, that compromises accuracy in cases when the first model prediction had errors.

**GPU memory.** We reduced the peak GPU memory consumption in the inference pipeline with various simple code optimizations, including releasing all intermediate GPU memory variables right away and re-using GPU memory (e.g. softmax can be computed in-place). We switched to non-overlapping sliding window inference and copied to GPU memory only the current window patch, and transferred the results to CPU immediately, instead of keeping full input and outputs on GPU. This inevitably led to a compromise between compute time and peak GPU memory. Ultimately, we decided to prioritize peak GPU memory, and keep it under 4GB, and allow computations to take longer. It was easier to control, unlike the overall compute time which depends on several factors (including the input image size).

**Left-right merging** One small optimization we attempted was to merge labels with left-right symmetry into a single class, specifically merging both kidneys and both adrenal glands classes. The output of the network becomes 13 (a background, 11 organs and 1 tumor class), instead of 15. This saves a bit of GPU memory, as the network output is smaller. Another reason for merging, was our hypothesis that it will be easier for the model to learn the organ segmentation without attempting to differentiate left-right symmetry. At inference, a simple heuristic was used to divide merged labels into the left and right counterparts, based on the aorta center-line and two largest components of the merged kidney classes.

**1D connected component analysis** For post-processing, we used a simplified 1D connected component analysis (see Sec. 2.6)

**Unused optimizations** . Several optimizations we tried, but ultimately not used. Compiling the model with torch.compile() did lead to a faster inference, but at a price of several seconds required for compilation. Since the overall timing is measured on each case independently after a cold start, we decided against compilation. We also considered training models at a lower resolution of $1.3 \times 1.3 \times 1.3 \text{mm}^3$, which reduced image size and increased the overall inference time, at a price of segmentation accuracy. And finally, it was possible to forgo ensembling of 5 models completely, and simply use 1 model, which drastically improved the computational time, but again at the price of the segmentation accuracy.

### 2.6   Post-processing

We used several simplified post-processing techniques, since the overall computational time was one the FLARE23 metrics. We remove the tumor class predictions of less than a 100 voxels total. We also keep the largest connected component for a subset of organs (liver, kidneys, spleen, pancreas, aorta). Furthermore, for each of these organs, we ran a connected component analysis in 1D instead of a 3D analysis (the image is converted into a vector with ones for slices with at least one foreground voxel). This simplified 1D approach is able to remove only some disconnected inf/sup outliers (but not axial in-plane outliers).

## 3   Experiments

### 3.1   Dataset and evaluation measures

The FLARE 2023 challenge is an extension of the FLARE 2021-2022 [14][15], aiming to promote the development of foundation models in abdominal disease analysis. The segmentation targets cover 13 organs and various abdominal lesions. The training dataset is curated from more than 30 medical centers under

the license, including TCIA [3], LiTS [1], MSD [20], KiTS [8,9], autoPET [5,4], TotalSegmentator [23], and AbdomenCT-1K [16]. The training set includes 4000 abdomen CT scans where 2200 CT scans with partial labels and 1800 CT scans without labels. The validation and testing sets include 100 and 400 CT scans, respectively, which cover various abdominal cancer types, such as liver cancer, kidney cancer, pancreas cancer, colon cancer, gastric cancer, and so on. The organ annotation process used ITK-SNAP [24], nnU-Net [11], and MedSAM [13].

The evaluation metrics encompass two accuracy measures—Dice Similarity Coefficient (DSC) and Normalized Surface Dice (NSD)—alongside two efficiency measures—running time and area under the GPU memory-time curve. These metrics collectively contribute to the ranking computation. Furthermore, the running time and GPU memory consumption are considered within tolerances of 15 seconds and 4GB, respectively.

### 3.2  Implementation details

**Environment settings** The development environments used for training is presented in Table 2, and was done inside of a docker "nvidia/pytorch:23.06-py3", which comes with PyTorch 2.1 and many libraries preinstalled.

**Table 2.** Development environments and requirements.

| | |
|---|---|
| Docker | nvcr.io/nvidia/pytorch:23.06-py3 |
| System | Ubuntu 22.04.2 LTS |
| CPU | Intel(R) Xeon(R) Platinum 8362 CPU |
| RAM | 950G |
| GPU (number and type) | 8x NVIDIA A40 48G |
| CUDA version | 12.1 |
| Programming language | Python 3.10 |
| Deep learning framework | MONAI 1.2, PyTorch 2.1 |

**Training protocols** 1. The training was done in 2 stages (see 2.3). For each stage we trained 5 models (based on a random 5-fold split), and kept the best checkpoint based on the corresponding validation Dice value. The first stage checkpoints were used only to append pseudo-labels to the cases with missing labels (partial labels). After that 5 more models were trained on the full set of 2200 cases.

2. We use several augmentations including random rotation and scale (in axial plane only), random flips, random histogram shift and random contrast adjustment from MONAI transforms.

3. We crop 1 random patch per image of size $224{\times}224{\times}224$ voxels (which is equivalent to a batch size of 8 on 8 GPU machine). The crop is sampled with equal probability from the one of the 15 regions (background, tumor and 13 organs)

4. We use only the Dice value (on the validation fold subset) to select the best checkpoint.

**Table 3.** Training protocols.

| | |
|---|---|
| Network initialization | Random |
| Batch size | 8 |
| Patch size | 224×224×224 |
| Total epochs | 300 |
| Optimizer | AdamW |
| Initial learning rate (lr) | 2e-4 |
| Lr decay schedule | Cosine |
| Training time | 24 hours (per 1 model) |
| Loss function | dice-focal loss |
| Number of model parameters 87M | |

## 4    Results and discussion

We trained 5 models using 5-fold cross-validation on the 2200 cases (with added pseudo labels). Based on our random 5-fold split, the average Dice scores per fold per class are shown in Table 4. When computing the Dice score, for each class, only the original ground truth labels were used and the missing labels were skipped. We obtained a good organ segmentation accuracy, with many organs having average Dice scores above 95%, and a low tumor Dice score of 44.71% on average.

The low Dice score of the tumor class can be attributed to several factors. A large sub-set of the FLARE23 dataset included only the tumor class labels (888 cases out of 2200, see 2.3). Many of these cases come from the Autopet [5,4] challenge dataset, whose goal is a whole body tumor segmentation from paired 3D PET/CT images. The ground truth Autopet labels were primarily contoured on the PET modality using the hyper-intensity indicators. If we understand it correctly, the FLARE23 dataset adopted only the 3D CT portion of these images. So, there may not be enough information in CT images alone to identify tumors. Secondly, we hypothesize, that since many FLARE23 data subsets come from *organ* focused challenges, they may have missing tumor labels (e.g., a labeled liver class, without labeling tumors within the liver). This would create a conflicting ground truth for the algorithm to differentiate tumors and organs.

FLARE23 evaluation metrics assign a high value to tumor: equal value to the tumor accuracy vs all the organs combined. In effort to improve the tumor Dice score, we tried to retrain with a re-weighted loss. Instead of equal weighted averaging of a loss value per class, we assign a high weight (e.g. 10) to the tumor loss component. We were able to improve the tumor class average Dice score to

~50% (up from 44.71%), at the price of reducing average organs Dice score by almost ~10%. Visually, these results looked worse for organ segmentation, and the tumor segmentation improvements were difficult to judge. Since the ground truth tumor labels were the least consistent/reliable in the training dataset (compared to the organs labels), we ultimately decided not to use prioritize tumor loss, and not to use this strategy.

**Table 4.** Dice accuracy per class using our 5-fold training data random split. Each fold corresponds to the best checkpoint model trained during 5-fold cross-validation.

|  | fold 0 | fold 1 | fold 2 | fold 3 | fold 4 | Average |
|---|---|---|---|---|---|---|
| Liver | 97.08 | 97.18 | 97.19 | 97.29 | 97.39 | 97.22±0.11 |
| Right Kidney | 95.73 | 94.77 | 94.98 | 95.21 | 95.38 | 95.21±0.33 |
| Spleen | 96.59 | 96.95 | 97.04 | 96.94 | 96.87 | 96.87±0.15 |
| Pancreas | 85.41 | 85.93 | 86.31 | 86.96 | 86.36 | 86.19±0.51 |
| Aorta | 96.57 | 96.39 | 96.44 | 96.76 | 96.70 | 96.57±0.14 |
| Inferior vena cava | 95.18 | 94.73 | 95.71 | 95.00 | 94.74 | 95.07±0.36 |
| Right adrenal gland | 88.12 | 89.90 | 90.09 | 89.71 | 87.54 | 89.07±1.03 |
| Left adrenal gland | 87.80 | 89.23 | 90.16 | 89.83 | 87.39 | 88.88±1.09 |
| Gallbladder | 94.79 | 90.31 | 93.38 | 94.71 | 93.11 | 93.26±1.62 |
| Esophagus | 90.61 | 90.81 | 91.94 | 91.26 | 90.63 | 91.05±0.50 |
| Stomach | 96.75 | 96.77 | 97.33 | 97.37 | 96.65 | 96.97±0.31 |
| Duodenum | 91.48 | 91.24 | 92.93 | 93.06 | 89.75 | 91.69±1.21 |
| Left kidney | 95.77 | 95.11 | 94.76 | 95.10 | 95.59 | 95.26±0.37 |
| Tumor | 43.51 | 41.54 | 43.69 | 48.30 | 46.49 | 44.71±2.39 |
| Average |  |  |  |  |  | 89.55±12.96 |

The compute performance of our submission docker is shown in Table 4. We were able to achieve a peak GPU memory allocation of 3.5GB (well below the recommended 4GB minimum). Furthermore, the peak GPU memory allocation is independent of the input size. This allows our method to run even on low GPU memory GPU, or on a shared environment where GPU resources are shared between several applications. The average compute time of ~25 seconds was above the recommended 15 seconds minimum. It was possible to reduce it by e.g. using only a single model (instead of a 5 model ensemble), but we decided to keep the ensemble to maintain better segmentation quality. Furthermore, about a half of the run-time was used to start the docker and load the libraries and the model checkpoints for each input image from scratch. In practice these steps can be done just once, and the overall inference time per case becomes much smaller.

### 4.1   Qualitative results on validation set

A visualization of the ground truth labels and the predicted results of one of the validation cases is shown in Fig. 2. Organs are accurately segmented inline with Tab. 4 results, with a slight undersegmentation of the pancreas class. The tumor

**Table 5.** Quantitative evaluation of segmentation efficiency in terms of the running them and GPU memory consumption. Total GPU denotes the area under GPU Memory-Time curve. Evaluation GPU platform: NVIDIA QUADRO RTX5000 (16G) CUDA 11.8. The numbers are provided by the organizers after running our docker.

| Case ID | Image Size | Running Time (s) | Max GPU (MB) | Total GPU (MB) |
|---|---|---|---|---|
| 0001 | (512, 512, 55) | 23.5 | 3562MB | 470995 |
| 0051 | (512, 512, 100) | 26.2 | 3562MB | 692182 |
| 0017 | (512, 512, 150) | 20.2 | 3562MB | 392637 |
| 0019 | (512, 512, 215) | 25.5 | 3562MB | 619403 |
| 0099 | (512, 512, 334) | 29.7 | 3562MB | 682317 |
| 0063 | (512, 512, 448) | 28.7 | 3562MB | 647214 |
| 0048 | (512, 512, 499) | 22.68 | 3562MB | 598048 |
| 0029 | (512, 512, 554) | 21.61 | 3562MB | 532494 |

segmentation includes a cut-off undersegmentation artifact, most likely caused by the sliding window inference with no overlap (we removed the default overlap of 0.625 as a trade off to speed up the inference).

### 4.2 Limitation and future work

Some limitations of the current work include not using the unlabeled data, and using a simple pseudo-labeling method. A dedicated semi-suprvised or unsupervised learning method, could provide better accuracy. Future work can include a better technique to handle tumor class segmentation, taking into account its diverse variability in various data subsets.

## 5 Conclusion

In this work, we describe our method submission to the FLARE23 challenge, using Auto3DSeg from MONAI. Our method is a supervised semantic segmentation, which uses pseudo-labels on the missing classes, to be able to work with partially labeled data. Our method achieves Dice scores of 93% for organs classes, and 43% for the tumor class.

**Acknowledgements** The authors of this paper declare that the segmentation method they implemented for participation in the FLARE 2023 challenge has not used any pre-trained models nor additional datasets other than those provided by the organizers. The proposed solution is fully automatic without any manual intervention. We thank all the data owners for making the CT scans publicly available and CodaLab [19] for hosting the challenge platform.

We also thank FLARE23 organizers for assembling such a large combined dataset, and for creating this valuable platform for researchers to compare and evaluate their methods.

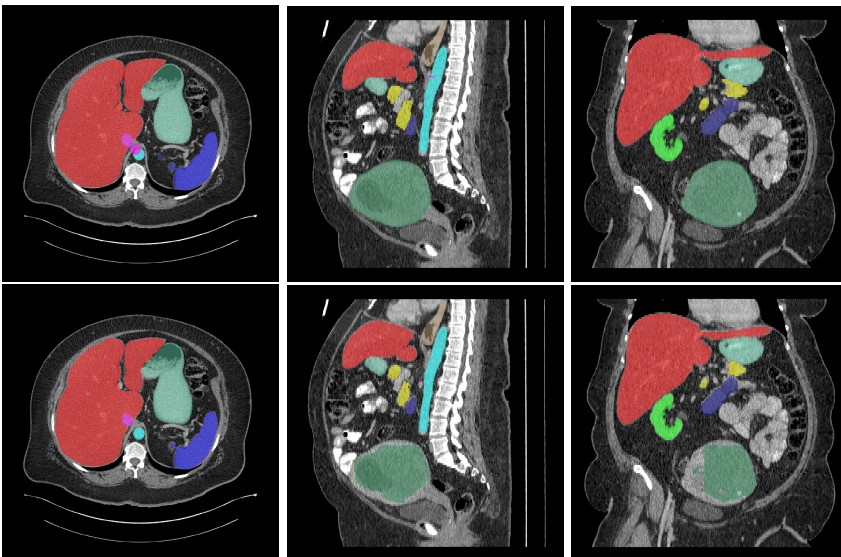

**Fig. 2.** A visualization of the FLARE23Ts_0017 case (axial slice 117): top row - ground truth, bottom row - predicted result. Visually organ segmentation matches the groundtruth well, with our predicted result slightly under-segmenting Pancreas (in yellow). Interestingly, the ground truth label incorrectly includes 2 inferior vena cavas (in pink/magenta) on this slice, with one of them even overlapping with aorta (in cyan). A large tumor in the lower abdomen was accurately detected but substantially under-segmented in our prediction result (in dark green). Straight edges of the tumor cut-off indicate that one of the reasons for under-segmentation could be the sliding window inference without overlap (which produces this boundary artifact).

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

**Table 6.** Checklist Table. Please fill out this checklist table in the answer column.

| Requirements | Answer |
|---|---|
| A meaningful title | Yes |
| The number of authors ($\leq$6) | 4 |
| Author affiliations and ORCID | Yes |
| Corresponding author email is presented | Yes |
| Validation scores are presented in the abstract | Yes |
| Introduction includes at least three parts: background, related work, and motivation | Yes |
| A pipeline/network figure is provided | Fig. 1 |
| Pre-processing | Sec. 2.1 |
| Strategies to use the partial label | Sec. 2.3 |
| Strategies to use the unlabeled images. | Sec. 2.3 |
| Strategies to improve model inference | Sec. 2.5 |
| Post-processing | Sec. 2.6 |
| Dataset and evaluation metric section is presented | Sec. 3.1 |
| Environment setting table is provided | Tab. 2 |
| Training protocol table is provided | Tab. 3 |
| Efficiency evaluation results are provided | Tab. 4 |
| Limitation and future work are presented | Yes |
| Reference format is consistent. | Yes |