# OpenReview forum: "Automated segmentation of organs and tumors from partially labeled 3D CT in MICCAI FLARE 2023 Challenge."
_MICCAI.org/2023/FLARE — Submitted to FLARE 2023_

### Official Review · Reviewer_P3j1 · 2023-10-25
**Automated segmentation of organs and tumors from partially labeled 3D CT in MICCAI FLARE 2023 Challenge**

**Rating:** 7
**Confidence:** 5

**Review:**

The title should be more meaningful.
Please include the DSC of organs and lesions, the average running time,  and the area under GPU memory-time curve on the public validation set.
Table 4 should follow the format in the template.

---

### Official Review · Reviewer_EX99 · 2023-10-25
**great work**

**Rating:** 8
**Confidence:** 5

**Review:**

#Reviewer 3

The paper is well-written! Really nice work!

1. Please report the average organ and tumor DSC score. The online validation LB is still open for submissions.
2. Please indicate the method in the title (i.e., SegResNet)
3. It would be great if you could mark the shapes in each block in the network architecture figure.

#Reviewer 4

1. The average running time and area under GPU memory-time curve are missing in the Abstract section.
2. The font in the picture should be Times New Roman.
3. It would be great if you could add ablation studies to analyze the effect of unlabelled data.
4. Please show at least two examples with good segmentation results and two examples with bad segmentation results in the validation set.

---

### Public Comment · ~PENGJU_LYU1 · 2023-11-26

add public validation and test result

---

### Decision · Program_Chairs · 2023-10-25

**Decision:**

Reject

**Comment:**

The authors didn't make responses to the valuable review comments.